# Effects of 4-Pentenoic Acid and Malic Acid on Dynamics of Bacterial Communities and Fermentation Characteristics in Nettle Silage

**DOI:** 10.3390/microorganisms13092088

**Published:** 2025-09-07

**Authors:** Yingchao Sun, Yuxin Chai, Shuangming Li, Yongcheng Chen, Chunhui Ma, Rongzheng Huang, Fanfan Zhang

**Affiliations:** College of Animal Science & Technology, Shihezi University, Shihezi 832000, China; 17633866730@163.com (Y.S.); 17799931814@163.com (Y.C.); lishuangming0410@163.com (S.L.); chenyonchn@163.com (Y.C.); chunhuima@126.com (C.M.)

**Keywords:** nettle, 4-pentenoic acid, malic acid, silage, bacteria

## Abstract

Background: Nettle is a potential non-conventional feed resource due to its high level of crude protein content, and ensiling is better for utilization in the animal industry. Previous integrated analysis (microbiome and metabolome) suggested that 4-pentenoic acid and malic acid in ensiled nettle may inhibit harmful microorganisms within the system. The present study investigated the effects of these two acids on nettle silage quality through the addition of 1% fresh weight of these acids, then analyzed the characteristics and bacterial communities during 60 days of nettle ensiling. Results: The addition of 4-pentenoic acid increased the content of crude protein (CP) and acetic acid (AA) compared with both control and malic acid-treated groups during 30 to 60 days of ensiling (*p* < 0.05). Lactic acid (LA) content was highest in the malic acid-treated group (4.21%, dry matter, DM based) compared to the control and 4-pentenoic acid-treated groups after 7 days of nettle ensiling (*p* < 0.05), but lower compared with the 4-pentenoic acid-treated group after 30 days of nettle ensiling (*p* < 0.05), and it was not detected in all groups after 60 days of silage. The contents of butyric acid (BA) and ammonia (AN) were the lowest (2.92–4.39% of DM and 9.94–24.28% of total nitrogen, respectively) in the 4-pentenoic-treated group compared with both control and malic acid-treated groups during 30 to 60 days of ensiling (*p* < 0.05). Both acids increased the relative abundance of *Weissella* after 30 days of nettle ensiling, with 4-pentenoic acid showing a higher inhibitory capacity. Both acids showed a trend to inhibit the relative abundance of *Clostridium sensu stricto 15* after 30 days of nettle ensiling. *Clostridium sensu stricto 15* showed significant positive correlation with BA and AN (*p* < 0.05). Conclusions: The results of the present study suggested that the addition of 4-pentenoic could improve the quality of silage by reducing levels of protein degradation, probably resulting from its inhibited activity against *Clostridium* spp. However, malic acid was less effective than 4-pentenoic acid in suppressing *Clostridium* spp. activity and the associated production of BA and AN, resulting in inferior preservation of CP.

## 1. Introduction

The rapid expansion of the livestock feed industry in recent years has caused critical shortages of conventional protein sources (e.g., corn and soybean meal), which now fail to meet market demands. Developing non-conventional feed sources is now an imperative necessity for ensuring sustainable livestock production systems [1].

Nettles (*Urtica* spp.) are annual or perennial herbaceous plants belonging to the family Urticaceae. Characterized by high nutritional value, robust environmental adaptability, and strong stress resistance, they exhibit a widespread global distribution [2]. Furthermore, nettles are rich in essential nutrients including proteins, carbohydrates, lipids, and amino acids, as well as biologically active compounds such as carotenoids, terpenoids, α-linolenic acid, 12-linoleic acid, vitamins, cis-9, and polyphenolic compounds [3]. In addition, these plants possess demonstrate antioxidant, antibacterial, and anti-inflammatory properties, positioning them as a highly promising alternative feed resource. Recent research demonstrates the significant potential of nettle species as functional feed ingredients for optimizing animal production, enhancing stress resilience, and improving the quality of animal-derived products across livestock and poultry species. Specifically, partial alfalfa (*Medicago sativa*) replacement with nettles enhanced antioxidant status in dairy cows [4], dietary supplementation ameliorated heat stress in broilers [5], and inclusion improved egg yolk carotenoid deposition in laying hens [6]. Similarly, *Boehmeria nivea* hay substitution boosted rabbit productivity [7].

Similar to alfalfa, nettles exhibit constrained silage characteristics due to several intrinsic factors [8]. Their elevated crude protein content and high buffering capacity, coupled with a concomitantly low water-soluble carbohydrate (WSC) content, result in insufficient fermentable substrate, thereby restricting the proliferation of essential lactic acid bacteria (LAB) during the silage process [9]. Furthermore, the abundant protein serves as a readily available nitrogen source, promoting the growth of undesirable Clostridia species. Subsequent proteolysis by *Clostridium* spp. generates ammonia (AN), which neutralizes lactic acid (LA) and impedes the critical decline in system pH [10]. This disruption of the acidification process causes ineffective silage fermentation.

Our previous study showed that malic acid enhances silage *Lactobacillus* growth through specialized metabolic utilization and competitive dominance [3]. *Lactobacillus* spp. express *mleS* and *mic* genes encoding malolactic enzyme and malic enzyme, enabling direct decarboxylation of malic acid to LA via malolactic fermentation (MLF) [3]. Concurrently, malic acid is funneled into pyruvate metabolism (*pdhB*, *ldh* upregulation) and glycolysis, sustaining energy production and biosynthetic precursors [11]. Additionally, naturally occurring antimicrobial compounds within nettles, such as 4-pentenoic acid, formic acid, and phenolic compounds, exhibit broad-spectrum inhibitory activity against microbial growth. 4-Pentenoic acid may act as a potential spoilage factor in nettle silage, as it demonstrated inhibitory effects against beneficial LAB (e.g., *Pediococcus pentosaceus*) in vitro through impairing fatty acid synthesis and cell membrane integrity, which could contribute to the observed quality deterioration [11]. Whereas prior microbiome analyses identified 4-pentenoic acid and malic acid as key fermentation inhibitors in nettle silage [3,11], their targeted application as antibacterial additives against *Clostridium* remains unexplored. To address this knowledge gap, we added these two acids in nettle silage in order to determine how these acids are involved in the fermentation process. We hypothesized that exogenous addition of 4-pentenoic acid—an endogenous antimicrobial metabolite in nettle silage—would selectively inhibit proteolytic *Clostridium* spp., thereby reducing protein degradation and ammonia production, while malic acid would primarily enhance lactic acid bacterial activity via metabolic utilization, offering a comparative perspective on acid-mediated preservation strategies. The specific objectives of this study were to evaluate the effects of 4-pentenoic acid and malic acid on the fermentation characteristics of nettle silage, analyze their ensuing impacts on the bacterial community dynamics with a focus on the suppression of spoilage-associated *Clostridium sensu stricto 15* and the enrichment of beneficial genera, and correlate these microbial shifts with fermentation parameters to elucidate the mechanistic basis of acid-mediated preservation in high-protein silages.

## 2. Materials and Methods

### 2.1. Silage Preparation

Nettle (*Urtica cannabina*) was harvested on 10th September 2023 from the wild in Shawan County, central mountains of Tianshan, Xinjiang, China (84°58′–86°24′ E; 43°26′–45°20′ N). After air-drying to 394.2 g·kg^−1^ fresh weight, plants were cut into 2 cm segments using a fodder chopper. Approximately 1.0 kg of homogenized material was vacuum-sealed in polyethylene bags (23 cm × 30 cm) fitted with one-way valves. Two treatments were established, the addition of 1% of fresh weight for malic acid (MA, BioUltra grade, Macklin Biochemical Co., Ltd., Shanghai, China; Product No. M815619) and 4-pentenoic acid (4PA, ≥97% purity, Aladdin Biochemical Technology Co., Ltd., Shanghai, China; Product No. P112354), respectively, based on our previous research demonstrating the efficacy of 1% of fresh weight in modulating microbial activity and fermentation in nettle silage, particularly for suppressing undesirable Clostridia [11]. Control groups were also prepared. Each treatment comprised nine replicates, totaling 45 sample bags stored at 24 °C for 60 days.

### 2.2. Characteristics Analysis of Nettle Silage

After 7, 30, and 60 days of ensiling, 200 g silage samples were dried at 65 °C for 48 h and ground to pass through a 1.0 mm sieve, and then used to determine DM content. Total nitrogen (TN) was determined using an automatic Kjeldahl nitrogen analyzer (K9840, Hanon Co., Ltd., Qingdao, China), and crude protein was calculated according to the method of the Association of Official Agricultural Chemists (AOAC). WSC content was determined according to methods described above [12].

A silage sample of 20 g from each treatment was combined with 180 mL of deionized water to assess fermentation characteristics. The mixture (1:9 for *v*/*v*) was filtered through four layers of cheesecloth. The pH was immediately measured with a portable pH meter (PHS-3C, Instrument and Electrical Science Instrument Co., Ltd., Shanghai, China), and the supernatant was retained for ammonia and organic acid analyses following established protocols [3,13]. Organic acids (actic acid, LA; acetic acid, AA; propionic acid, PA; butyric acid, BA) were quantified via HPLC employing a C18 column (150 × 4.6 mm, 94 FMF-5559-EONU, FLM Scientific Instrument Co., Ltd., Guangzhou, China). Na_2_HPO_4_ (1 mM, pH 2.7) and methanol were used for mobile phase A and B, respectively, with a flow rate of 0.6 mL·min^−1^. Separation was performed using a 20 μL injection volume at 50 °C with the following gradient: 0 min, 100% A; 5–40 min, 98% A.

### 2.3. Sequencing Analysis of Bacterial Communities in Nettle Silage

Total DNA was extracted from triplicate samples per treatment group after 7, 30, and 60 days of ensiling using the commercial DNA Kit (FastDNA^®^ Spin Kit for Soil, MP Biomedicals, Irvine, CA, USA). Primers (338F: ACTCCTACGGGAGGCAGCAG; 806R: GGACTACHVGGGTWTCTAAT) targeting the V3–V4 regions of 16S rDNA were used for PCR amplification. The amplicons were extracted, purified, and the raw sequences analyzed [3]. The sequences were uploaded to the National Center for Biotechnology Information (NCBI, https://www.ncbi.nlm.nih.gov): PRJNA1233857 (accessed on 13 March 2025) for addition with 4-pentenoic acid and PRJNA1236718 (accessed on 16 March 2025) for addition of malic acid into nettle silage, respectively.

### 2.4. Statistical Analysis

The effects of acid additives on fermentation characteristics and bacterial communities were assessed within each sampling time point (7, 30, and 60 days) to evaluate treatment efficacy at distinct stages of the ensiling process. Data from the three silage groups underwent one-way ANOVA in SPSS 22 (SPSS Inc., Chicago, IL, USA), with Tukey’s test (*p* < 0.05) determining treatment differences. Microbiota bioinformatics utilized the Majorbio Cloud platform (accessed on 10 March 2025. https://cloud.majorbio.com). Pearson correlation analysis was conducted to assess relationships between bacterial genera (relative abundance > 1%) and fermentation parameters using the “corrplot” package in R (v4.3.1). Correlation coefficients and significance (*p* < 0.05) are reported, with visualization via heatmap.

## 3. Results

### 3.1. Effects of 4-Pentenoic Acid and Malic Acid on Characteristics of Nettle Silage

The characteristics of nettle after wilting were 39.42% fresh weight of dry matter, 16.89% DM of crude protein, 7.50% DM of WSC, and 8.04 of pH. As shown in Table 1, DM was highest in the MA-treated group, compared to the control and 4PA groups after 30 days of nettle ensiling (*p* < 0.05). CP was the lowest in the malic acid-treated group compared to the control and 4-pentenoic acid-treated group after 7 days of nettle ensiling (*p* < 0.05) and the highest in the 4PA-treated group, followed by control during 30–60 days of nettle ensiling. The 4PA-treated group showed the highest pH value compared with control and MA-treated groups after 7 days of nettle ensiling (*p* < 0.05), while it showed the lowest pH after 30–60 days of ensiling (*p* < 0.05).

As shown in Table 1, DM was highest in the MA-treated group compared with control and 4PA-treated groups after 30 days of ensiling (*p* < 0.05). The content of CP was higher in the 4PA-treated group compared to the MA-treated group after 7 days of ensiling (*p* < 0.05), showed highest content in the 4PA-treated group compared to the other two groups during 30–60 days of ensiling (*p* < 0.05). The content of WSC showed no difference between control and each two treated groups during entire ensiling process (*p* > 0.05). The pH value (7.98) was highest in 4PA-treated group after 7 days of ensiling, but lower during 30–60 days of ensiling compared with control and MA-treated groups (*p* < 0.05). The content of LA was highest in the MA-treated group after 7 days of ensiling, compared to the control and 4PA-treated groups (*p* < 0.05), but lower after 30 days of ensiling compared to the 4PA-treated group (*p* < 0.05), and lowest in control after 30 days of ensiling compared to the other two treated groups (*p* < 0.05). The content of AA was highest in the 4PA-treated group during 30–60 days of ensiling compared to the control and MA-treated groups (*p* < 0.05). The content of PA was highest in the control compared with two treated groups after 60 days of ensiling (*p* < 0.05). The content of BA was lowest in the 4PA-treated group during 30–60 days of ensiling compared to the control and MA-treated groups (*p* < 0.05). Ammonia was lower in the 4PA-treated group compared to the control and MA-treated groups (*p* < 0.05) after 30 days of ensiling, and lowest in the 4PA-treated group compared to the control and MA-treated groups (*p* < 0.05) after 60 days of ensiling.

### 3.2. Effect of 4-Pentenoic Acid and Malic Acid on Bacteria Community of Nettle Silage

As shown in Figure 1A,B, both chao 1 and ace index showed no difference among each treated group and control (*p* > 0.05). The Shannon index was highest in the MA-treated group after 60 days of nettle ensiling compared with 7 days (*p* < 0.05, Figure 1C).

As shown in Figure 2, there is a clear difference between the control and the acid-treated groups after 30 days of nettle ensiling.

As shown in Figure 3A, Firmicutes were the most dominant bacteria in control and each treated group, followed by Cyanobacteria and Proteobacteria during 60 days of nettle ensiling, and no significant difference was observed among the groups (*p* > 0.05).

As shown in Figure 3B, *Aerococcus* (50.62–61.52%) were the dominant bacteria in both the control and two acid-treated groups after 7 days of nettle ensiling, followed by *Enterococcus* (5.66–12.32%) and *Jeotgalibaca* (2.51–9.25%), no significant difference was observed for these bacteria among the groups (*p* > 0.05). *Aerococcus* (20.56–41.07%) were the dominant bacteria in both the control and two acid-treated groups after 30 days of nettle ensiling, followed by *Irregularibacter* (2.50–39.78%) and *Atopostipes* (8.04–11.67%); no difference was observed for these bacteria among the groups (*p* > 0.05). *Desemzia* showed the significant highest relative abundance in the 4PA-treated group compared to the control and MA- treated grouops, with a higher level in the control group than in the malic acid-treated group after 30 days of nettle ensiling (relative abundances were 3.50%, 2.43%, and 4.16% for the control, MA-treated and 4-pentenoic acid-treated groups, respectively, *p* < 0.05). *Weissella* showed significant highest relative abundance in the 4PA-treated group compared to the control and MA-treated groups, with MA-treated group showing highest abundance than control after 30 days of nettle ensiling (relative abundances were 0.64%, 1.41%, and 2.55% for the control, MA-treated, and 4-pentenoic acid-treated groups, respectively, *p* < 0.05). Aerococcus (22.94–24.62%) were the dominant bacteria in both the control and two acid-treated groups after 60 days of nettle ensiling, followed by *Atopostipes* (15.53–22.45%) and *Irregularibacter* (9.37–21.30%); no difference was observed for these bacteria among the groups (*p* > 0.05). *Tissierella* showed significant highest relative abundance in the control compared to other two treated groups; the MA-treated group had higher relative abundance compared to the 4PA-treated group after 60 days of nettle ensiling (relative abundances were 6.85%, 2.30%, and 0.08% for the control, MA-treated, and 4-pentenoic acid-treated group, respectively, *p* < 0.05).

As shown in Figure 4, *Jeotgalibaca, Lactococcus,* and *Aerococcus* were the most significantly different bacteria in the control, 4-pentenoic acid- and MA-treated groups, respectively, after 7 days of nettle ensiling (*p* < 0.05). *Facklamia* and *Rubellimicrobium* were the most significantly different bacteria in the 4-pentenoic acid- and MA-treated groups, respectively, after 30 days of nettle ensiling (*p* < 0.05). *Atopostipes* and *Tissierella* were the most significantly different bacteria in the control and MA-treated groups, respectively, after 60 days of nettle ensiling (*p* < 0.05).

### 3.3. Correlation Analysis Between Bacteria Community and Fermentation Characteristics in Nettle Silage

As shown in Figure 5, *Tissierella*, *Anaerosporobacter*, *Irregularibacter, Tepidimicrobium*, Atopostipes, and *Clostridium sensu stricto 15* showed significant positively correlation with ammonia nitrogen, BA, and pH, while *Aerococcus*, *Lactococcus*, *Carnobacterium*, and *Enterococcus* showed significant negative correlation with each characteristics (*p* < 0.05). *Tissierella*, *Anaerosporobacter*, *Irregularibacter*, *Tepidimicrobium*, *Atopostipes*, and *Clostridium sensu stricto 15* showed significant negatively correlation with WSC, CP, LA, and AA, while *Aerococcus*, *Lactococcus*, *Carnobacterium*, and *Enterococcus* showed significant positive correlation with each characteristics (*p* < 0.05).

## 4. Discussion

### 4.1. Effect of 4-Pentenoic Acid and Malic Acid on Characteristics of Nettle Silage

Building on the finding that 4PA actively inhibits LAB activity [11], we specifically evaluate these plant-derived organic acids (4PA and MA) as targeted antimicrobial additives to suppress spoilage microorganisms, particularly *Clostridium* spp. This experimental design builds directly on previous work, which identified 4PA and MA as key limiting factors in nettle silage fermentation. Consequently, this work represents a mechanistically justified progression toward understanding acid-mediated preservation strategies in high-protein silages.

The highest DM content was attributed to malic acid after 30 days of ensiling, but this effect disappeared as the silage fermentation time progressed. After 60 days of ensiling, 4PA treatment reduced crude protein loss by 43% compared to control (CK: 21.4% DM loss vs. 4PA: 12.2% DM loss) and by 39.0% compared to malic acid (MA) treatment (2.85% DM loss), demonstrating superior preservation of protein integrity. The persistently high pH (>8.0 after 60 days) and the complete disappearance of LA in all groups (Table 1) align with established characteristics of failed nettle ensiling [3]. These phenomena were caused by clostridial dominance—*Clostridium sensu stricto 15* degrades proteins into AN, neutralizing organic acids and maintaining high pH. The accumulation of LA is primarily responsible for pH reduction in silage [14]. As present study showed, LA content decreased and eventually disappeared after 60 days of ensiling, which corresponded to changes in pH levels.

LAB mainly converts WSC into organic acids such as lactic acid and acetic acid [15]. Data revealed that content of WSC was stabilized during 30–60 days of nettle ensiling, indicating that LAB fermentation process probably stopped during this period. However, both ammonia and BA increased as the ensiling time progressed, indicating that Clostridial fermentation occurred during nettle ensiling. There are two types of *Clostridium* fermentation that cause silage to spoil and deteriorate—glycolysis such as *C. butyricum* that generates hydrogen, carbon dioxide, and BA through the fermentation of sugar and LA, and protein decomposition such as *C. sporogenes* that produces ammonium nitrogen and amines by breaking down free amino acids and proteins [16,17]. WSC content decreased by 72.26% after 30 days of nettle ensiling and stabilized as the ensiling time progressed. The same was observed for LA content, which eventually disappear with ensiling time prolonged to 60 days. The results probably suggest that these glycolysis types of *Clostridium* spp. prefer to utilize LA as a carbon source during nettle ensiling *C. sporogenes* indicated a strong capacity to degrade protein, especially when WSC is limited. As present study demonstrated, the content of ammonia increased 3-fold after 60 days of ensiling compared with 7 days of ensiling, indicating that the activity of protein decomposition type of *Clostridium* was significantly higher during ensiling.

4-pentenoic acid is not a conventional organic acid additive of silage; it is mainly found in nettle silage [3]. As a plant-specialized metabolite, 4-pentenoic acid uniquely targets proteolytic Clostridial in nettle silage—unlike conventional additives like malic acid. Little is known about the inhibiting activity of 4-pentenoic acid against bacteria; only one study observed that 4-pentenoic acid showed certain level of capacity to inhibit relative abundance of *C. botulinum* [18].

Our previous studies identified 4-pentenoic acid as an endogenous metabolite in naturally fermented nettle silage, exhibiting differential antibacterial activity: it strongly inhibits the beneficial lactic acid bacterium *Pediococcus pentosaceus* (MIC = 16 mg·mL^−1^) by disrupting cell membrane integrity and suppressing fatty acid biosynthesis (*fabG*) and acid-tolerant proteins (*accA*). Conversely, 4PA indirectly suppresses harmful bacteria (e.g., *Clostridium sensu stricto 15*) by promoting antagonistic LAB genera (e.g., *Enterococcus*), evidenced by significantly reduced AN production (*p* < 0.05) [3,11]. The present study supplemented 4-pentenoic acid at 1% fresh weight—a concentration exceeding typical endogenous levels (0.2–0.5 mg·g^−1^ DM) observed during natural fermentation. Results confirmed that exogenous 4-pentenoic acid significantly suppressed proteolytic *Clostridium sensu stricto 15*, reducing BA and AN production by 42.35% and 25.95%, respectively, after 60 days of ensiling. This demonstrates 4-pentenoic acid’s capacity to inhibit relative abundance of *Clostridium* spp., and particularly glycolytic types, though the pharmacological dosage used here may not reflect its physiological function in natural systems.

Malic acid effectively suppresses harmful microorganisms (e.g., Clostridial and molds) through rapid acidification, without significantly inhibiting beneficial LAB, while synergistically enhancing silage quality when combined with homofermentative LAB. As a conventional organic acid additive in silage production, studies demonstrate that malic acid supplementation at 0.6–1% (fresh weight basis) reduces AN content in silages with high CP levels (16.86–22.20% of dry matter, DM) [19,20,21]. Unexpectedly, in nettle silage (CP levels is 16.89% of DM), MA treatment increased AN production at the late stage of ensiling (day 60). This discrepancy likely stems from complex inhibitory interactions among endogenous biologically active compounds and microorganisms, necessitating further mechanistic investigation.

### 4.2. Effect of 4-Pentenoic Acid and Malic Acid on Bacteria Community of Nettle Silage

*Aerococcus* belong to *Lactobacillales*, often considered as “more peripheral” LAB, usually observed in fermented food [22]. Only a few studies have found that this bacterium exists in silage [23,24,25]. As present study results demonstrated, *Aerococcus* were the most dominant bacteria during entire nettle ensiling process, and acid treatments had no impact on it. The same result was observed in paper mulberry silage, where *Aerococcus* were dominant during ensiling, but dropped to 55.79% activity when inoculated with some strain of *Lactobacillus* [23]. However, the relative abundance of these bacteria was usually below 10% in alfalfa and pennisetum, and even below 1% after *Lactobacillus* was inoculated [24,25]. Clearly, *Lactobacillus* had the capacity to inhibit *Aerococcus* activity. Thus, the high relative abundance of *Aerococcus* was probably caused both nettle and paper mulberry containing a lot of bioactive compounds (polyphenol, flavonoids, organic acids, etc.) such as morin and naringin, which impact *Lactobacillus* activity [3,26].

Both acid treatments increased the relative abundance of *Weissella* after 30 days of nettle ensiling, with 4-pentenoic acid showing a higher capacity. *Weissella* is obligate heterofermentative LAB which initiates early fermentation of silage, and activity could be highly reduced when pH declines (usually < 5.0) with prolonged ensiling [27]. One study observed that malic acid decreased *Weissella* activity in the silage system, mainly due to pH rapidly declining and reaching 4.0 in *Moringa oleifera* leaf (MOL) silages [28]. The present study indicated that pH was over 8, suggesting that 4-pentenoic acid and malic acid probably were the main reason for the growth of this bacteria, but the specific mechanism need to be studied further. Furthermore, both acid treatments inhibited relative abundance of *Tissierella* after 60 days of nettle ensiling, with 4-pentenoic acid showing a higher inhibitory capacity. *Tissierella* could grow at pH range from 6.5 to 8.5 (optimum pH of 8.3) in sludge fermentation and produce butyric acid [29].

Additionally, the present study indicated that the 4PA-treated group showed a trend of inhibiting the relative abundance of *Clostridium sensu stricto 15* after 30 days of nettle ensiling. It is the first time that 4-pentenoic acid was found to show a capacity to inhibit *Clostridium* in silage. The same results were found for the addition of malic acid (5 g·kg^−1^ of FM) into alfalfa and cassava silage, proving it could decrease the relative abundance of *Clostridium* (BA and AN were significantly decreased.) [20,21].

Regarding the dual effects on LAB and Clostridia, both acids exhibited distinct patterns. MA transiently stimulated LAB activity early during ensiling (e.g., increased LA by 57.1% vs. control at day 7; Table 1), likely via its metabolic utilization by *Lactobacillus* [3,11]. However, this LAB-promoting effect diminished at day 30, coinciding with the resurgence of *Clostridium sensu stricto 15* (Figure 3B) and elevated BA/AN in MA silages. In contrast, 4PA initially suppressed LAB (e.g., reduced LA by 8.3% vs. MA at day 7), consistent with its inhibitory role against *Pediococcus* in nettle silage [3,11]. Crucially, 4PA demonstrated a superior and sustained inhibition of *Clostridium sensu stricto 15*, correlating with 42.4% lower BA and 26.0% lower AN vs. MA at day 60. This indicates that while both acids may partially inhibit LAB, 4PA’s selective and potent anti-clostridial activity (evidenced by its direct suppression of C. botulinum in vitro [11,18]) overrides its early LAB suppression, ultimately preserving protein integrity and fermentation quality. MA’s weaker clostridial inhibition likely stems from its primary role as a metabolic substrate rather than a dedicated antimicrobial agent.

Notably, based on the LEfSe analysis, the present study showed that *Jeotgalibaca* were potentially important bacteria involved in the early fermentation stage (7 days) of nettle silage; however, this shifted to *Lactococcus* and *Aerococcus* with the addition of malic acid and 4-pentenoic acid, respectively. *Facklamia* and *Rubellimicrobium* were potentially important bacteria involved in the middle-fermentation stage (30 days) of nettle silage following the addition of 4-pentenoic acid and malic acid, respectively. *Atopostipes* were potentially important bacteria involved in the later-fermentation stage (60 days) of nettle silage; however, this shifted to *Tissierella* following the addition of malic acid. Overall, the results suggested that both acids inhibited harmful microorganisms (such as *Clostridium*) throughout the entire fermentation period, with 4-pentenoic acid showing superior efficacy.

### 4.3. Correlation Analysis Between Bacteria Community and Fermentation Characteristics in Nettle Silage

The correlation analysis revealed significant relationships between bacterial genera and fermentation parameters, providing mechanistic insights into the acid-mediated preservation of nettle silage. *Clostridium sensu stricto 15* and *Tissierella* exhibited strong positive correlations with BA, AN, and pH, while showing negative correlations with WSC, CP, LA, and AA. This aligns with their known roles in proteolysis and saccharolytic fermentation, which lead to protein degradation, ammonia release, and butyrate accumulation—key indicators of silage spoilage [30].

In contrast, genera such as *Aerococcus*, *Lactococcus*, *Carnobacterium*, and *Enterococcus* demonstrated negative correlations with BA, AN, and pH, while positively correlating with WSC, CP, LA, and AA. Although *Aerococcus* dominated during the entire nettle ensiling process, its role in preserving CP and WSC (positive correlation) suggests potential metabolic activity that limited protein degradation. While *Aerococcus* showed positive correlation with CP, its persistent dominance in high-pH silage suggests environmental tolerance rather than proteolytic suppression [26]. Notably, *Lactococcus* and *Enterococcus*—typically associated with early-stage LAB fermentation—showed positive links to LA and AA but were insufficient to counteract clostridial bacteria activity in control groups [23].

Notably, the suppression of *Clostridium sensu stricto 15* and *Tissierella* by 4-pentenoic acid treatment—evidenced by a 98.8% reduction in Tissierella abundance—directly contributed to the observed decreases in BA (42.35%) and AN (25.95%) after 60 days of ensiling. This microbial inhibition activity in silage probably aligns with the mechanism of 4-pentenoic acid, which showed antibacterial activity in vitro [11]. Malic acid also reduced *Tissierella* (decreased to 66.42%) but was less effective against *Clostridium sensu stricto 15*, consistent with its weaker suppression of BA and AN production.

Furthermore, the enhanced abundance of *Weissella* in 4PA-treated silages correlated positively with AA accumulation, suggesting that this heterofermentative LAB may partially compensate for the lack of lactic acid production by contributing to acidification through acetic acid synthesis [31,32]. These findings collectively underscore the efficacy of 4-pentenoic acid in modulating the bacterial community to favor beneficial microbes and suppress proteolytic clostridia, thereby improving silage quality.

## 5. Conclusions

This study confirms that 4-pentenoic acid (addition of 1% FW) effectively enhances nettle silage quality by specifically inhibiting proteolytic *Clostridium sensu stricto 15*, thereby reducing BA and AN production, while preserving crude protein. Concurrently, it promotes *Weissella* activity and AA accumulation, partially compensating for LA deficiency. Compared to malic acid, 4-pentenoic acid demonstrates superior efficacy in controlling clostridial spoilage, establishing it as a novel promising plant-derived preservative for high-protein silages.

## Figures and Tables

**Figure 1 microorganisms-13-02088-f001:**
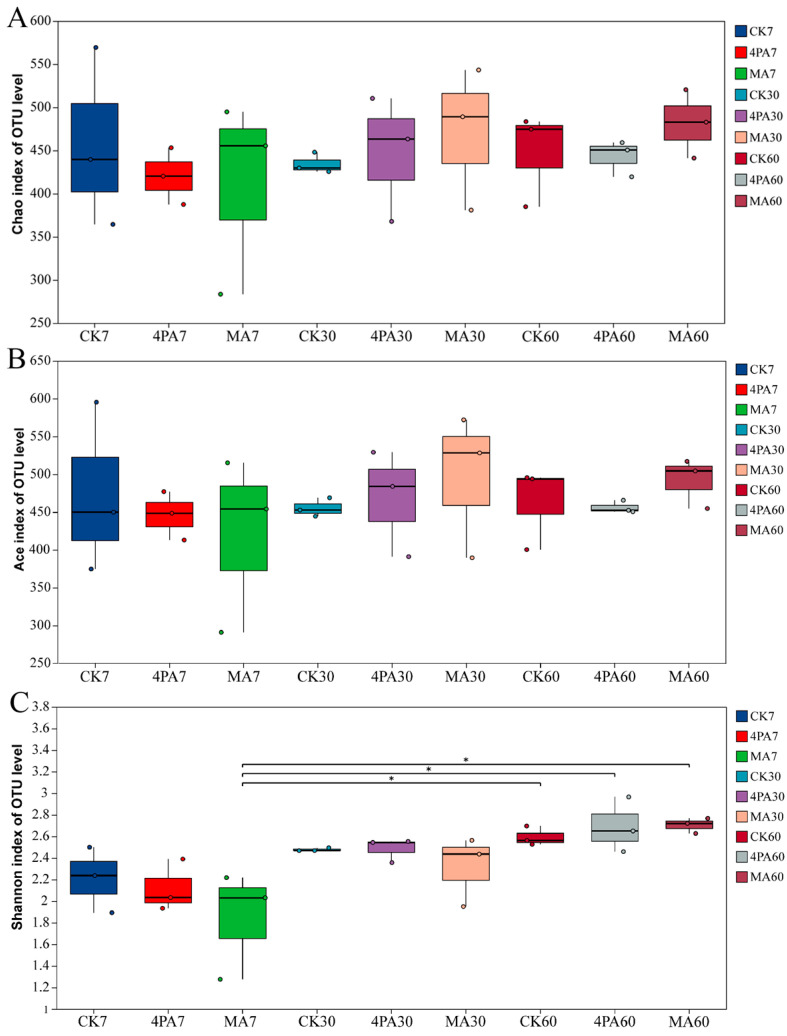
Alpha diversity of bacterial communities after addition of malic and 4-pentenoic acid during nettle ensiling. (**A**) Chao 1 index; (**B**) ace index; (**C**) Shannon index. CK7, 4PA7 and MA7: control group, addition of 4-pentenoic acid, and addition of malic acid after 7 days of nettle ensiling, respectively. CK30, 4PA30, and MA30: control group, addition of 4-pentenoic acid, and addition of malic acid after 30 days of nettle ensiling, respectively. CK60, 4PA60, and MA60: control group, addition of 4-pentenoic acid, and addition of malic acid after 60 days of nettle ensiling, respectively. “*” means *p* < 0.05. Same as below.

**Figure 2 microorganisms-13-02088-f002:**
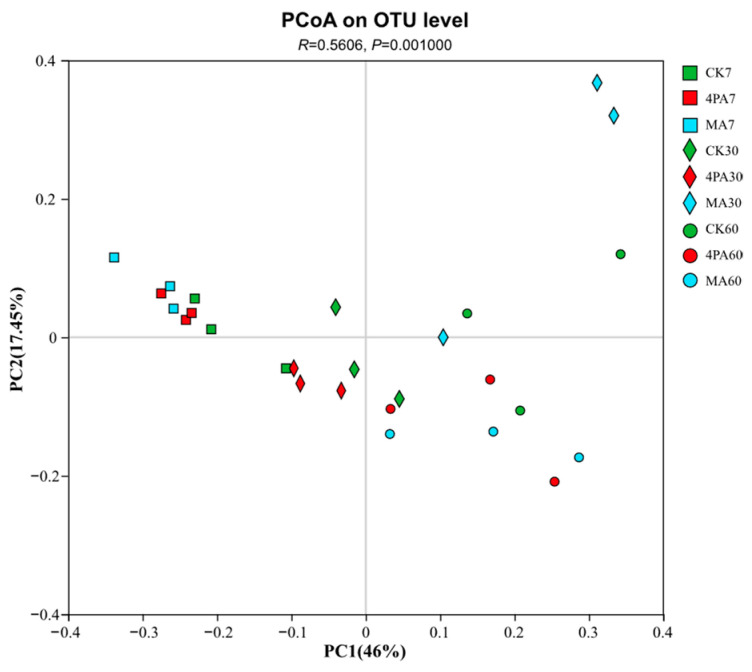
Principal coordinate analysis (PCoA) plots based on weighted UniFrac distance for bacterial communities and treatment (*R* = 0.5606, *p* = 0.00100). CK7, 4PA7, and MA7: control group, addition of 4-pentenoic acid, and addition of malic acid after 7 days of nettle ensiling, respectively. CK30, 4PA30, and MA30: control group, addition of 4-pentenoic acid, and addition of malic acid after 30 days of nettle ensiling, respectively. CK60, 4PA60, and MA60: control group, addition of 4-pentenoic acid, and addition of malic acid after 60 days of nettle ensiling, respectively.

**Figure 3 microorganisms-13-02088-f003:**
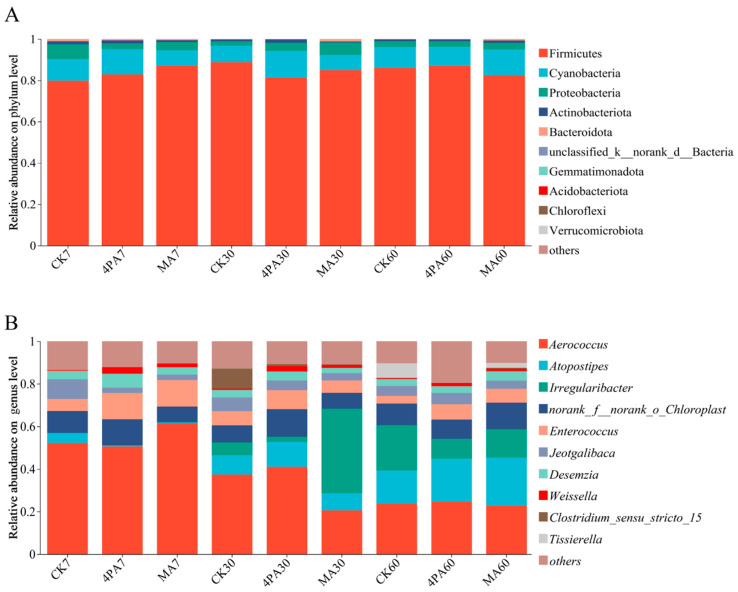
Bacterial community analysis at phylum (**A**) and genus levels (**B**) after addition of malic and 4-pentenoic acid during nettle ensiling. CK7, 4PA7, and MA7: control group, addition of 4-pentenoic acid, and addition of malic acid after 7 days of nettle ensiling, respectively. CK30, 4PA30, and MA30: control group, addition of 4-pentenoic acid, and addition of malic acid after 30 days of nettle ensiling, respectively. CK60, 4PA60, and MA60: control group, addition of 4-pentenoic acid, and addition of malic acid after 60 days of nettle ensiling, respectively.

**Figure 4 microorganisms-13-02088-f004:**
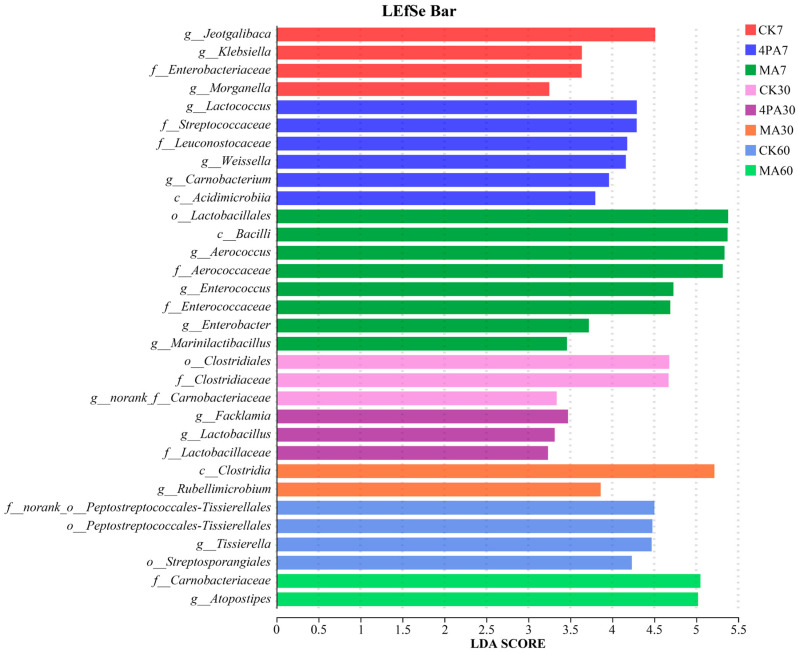
Comparison of microbial variations using LEfSe (LDA effect size) analysis of nettle silage after addition of malic and 4-pentenoic acid. CK7, 4PA7, and MA7: control group, addition of 4-pentenoic acid, and addition of malic acid after 7 days of nettle ensiling, respectively. CK30, 4PA30, and MA30: control group, addition of 4-pentenoic acid, and addition of malic acid after 30 days of nettle ensiling, respectively. CK60, 4PA60, and MA60: control group, addition of 4-pentenoic acid and addition of malic acid after 60 days of nettle ensiling, respectively.

**Figure 5 microorganisms-13-02088-f005:**
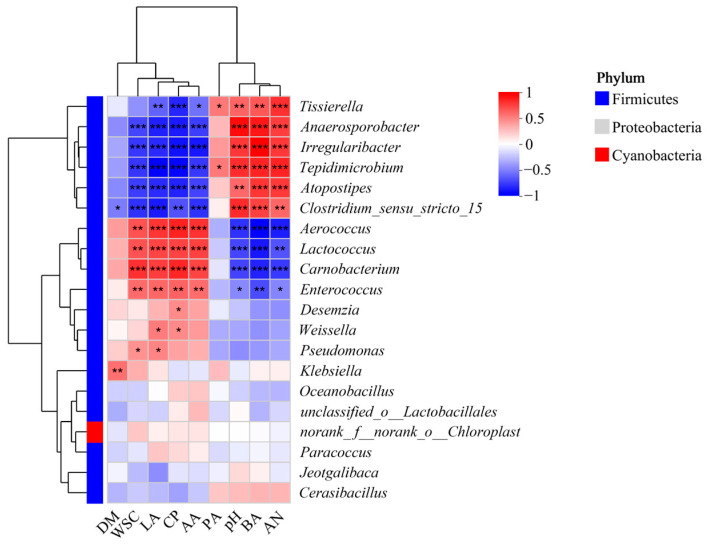
Heatmap of correlation analysis between bacteria and characteristic in nettle silage. Red/blue indicate positive/negative correlations. Correlations were computed using Pearson’s method with significance at *p* < 0.05. “*” means *p* < 0.05; “**” means 0.01 < *p* < 0.05; “***” means *p* < 0.001. Note: DM, dry matter; WSC, water-soluble carbohydrate; LA, lactic acid; CP, crude protein; AA, acetic Acid; PA, propanoic acid; BA, butyric acid; AN, ammonia.

**Table 1 microorganisms-13-02088-t001:** Characteristics during nettle ensiling after addition of relevant organic acids % DM.

Index	Time/Day	Groups			SEM	*p* Value
		CK	MA	4PA		
DM (% fresh weight)	7	37.72	37.23	36.58	1.245	0.675
	30	35.45 ^b^	36.66 ^a^	35.39 ^b^	0.038	0.027
	60	36.33	36.78	35.08	0.871	0.211
CP% DM	7	16.23 ^a^	15.89 ^b^	16.32 ^a^	0.039	<0.001
	30	15.27 ^c^	15.60 ^b^	15.96 ^a^	0.067	<0.001
	60	13.27 ^c^	13.72 ^b^	14.83 ^a^	0.062	<0.001
WSC% DM	7	2.92	3.30	2.95	0.230	0.263
	30	0.68	0.73	0.71	0.045	0.606
	60	0.69	0.74	0.68	0.041	0.386
pH	7	7.65 ^c^	7.80 ^b^	7.98 ^a^	0.024	<0.001
	30	8.35 ^a^	8.15 ^b^	8.11 ^b^	0.064	0.019
	60	8.47 ^a^	8.24 ^a^	8.10 ^b^	0.096	0.023
LA/(g·kg^−1^ DM)	7	2.68 ^b^	4.21 ^a^	3.86 ^b^	0.425	0.026
	30	0.016 ^c^	0.17 ^b^	0.40 ^a^	0.048	<0.001
	60	ND	ND	ND	-	-
AA/(g·kg^−1^ DM)	7	1.74	1.74	1.77	0.251	0.990
	30	0.94 ^b^	0.93 ^b^	1.57 ^a^	0.070	<0.001
	60	0.63 ^b^	0.50 ^b^	1.37 ^a^	0.166	0.004
PA/(g·kg^−1^ DM)	7	0.054	0.068	0.076	0.016	0.413
	30	0.082	0.078	0.050	0.019	0.215
	60	0.19 ^a^	0.083 ^b^	0.065 ^b^	0.026	0.005
BA/(g·kg^−1^ DM)	7	ND	ND	ND	-	-
	30	4.99 ^a^	5.59 ^a^	2.92 ^b^	0.287	<0.001
	60	8.31 ^a^	7.56 ^a^	4.79 ^b^	0.367	<0.001
AN (% of TN) DM	7	9.81	14.30	9.94	2.280	0.160
	30	12.90 ^a^	12.47 ^a^	10.86 ^b^	0.190	<0.001
	60	32.79 ^b^	39.33 ^a^	24.28 ^c^	1.150	<0.001

Note: CK: control group; MA: addition of 1% (fresh weight) malic acid; 4PA: addition of 1% (fresh weight) 4-pentenoic acid. Different lowercase letters in the same row indicate significant differences (*p* < 0.05). DM, dry matter basis; CP, crude protein; WSC, water-soluble carbohydrate; LA, lactic acid; AA, acetic acid; PA, propanoic acid; BA, butyric acid; AN, ammonia. Data are presented as the means ± standard error. ND: not detected. SEM: standard error of the mean.

## Data Availability

The datasets supporting the conclusions of this article are available in the National Center for Biotechnology (NCBI) Sequence Read Archive (SRA) of bacteria under accession numbers “PRJNA1233857” (accessed on 13 March 2025) for addition of 4-pentenoic acid and “PRJNA1236718” (accessed on 16 March 2025) for addition of malic acid into nettle silage, respectively. http://www.ncbi.nlm.nih.gov.

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
