# Peer review of "Effects of 4-Pentenoic Acid and Malic Acid on Dynamics of Bacterial Communities and Fermentation Characteristics in Nettle Silage"

_microorganisms, 2025, doi:10.3390/microorganisms13092088_

Round 1
Reviewer 1 Report (Previous Reviewer 2)
Comments and Suggestions for Authors
The authors have addressed all my comments, and the revised manuscript is now much clearer. I thank them for their thorough responses and improvements.
Author Response
We are deeply grateful to the Reviewer for their thoughtful and constructive comments, which have greatly enhanced the quality and clarity of our manuscript. We sincerely appreciate their time and effort in reviewing our work, and we are truly encouraged by their positive feedback. Their expertise and insightful suggestions were invaluable throughout the revision process.
Thank you once again for your generous acknowledgment and support. We wish the Reviewer continued success and all the best in their future endeavors.
Sincerely,
On behalf of all authors,
Yingchao Sun
Reviewer 2 Report (Previous Reviewer 3)
Comments and Suggestions for Authors
The article is very interesting, however, it cannot be published in its current form; it needs modifications.
1. What is the hypothesis of your work?
2. The wording of your objective is unclear.
3. Please carefully review the entire document. Some sections are highlighted in yellow, so I believe they have already been reviewed. If not, please remove the color.
4. Please review your statistical analysis because it does not describe all the analyses performed and their correlations.
5. I think you should consider the timeframe for your analyses and see if there are interactions by time and treatment.
Author Response
Thank you very much for taking the time to review our manuscript and offer valuable suggestions in your busy schedule. Your review is of great value to our research work.
For each of your suggestions, we have provided detailed individual responses below along with their positions (lines) in the revised manuscript. Meanwhile, in the resubmitted manuscript file, we have clearly marked all the corresponding revisions, corrections and modifications made based on your opinions in a highlighted (red) manner for your review.
We believe that the revisions made in response to the reviewers' comments have substantially enhanced the manuscript. We thank the reviewers once again for their constructive feedback, which has undoubtedly improved our work. We hope that the revised manuscript now meets the high standards of Microorganisms and is suitable for publication.
Sincerely,
On behalf of all authors,
Yingchao Sun

Round 2
Reviewer 2 Report (Previous Reviewer 3)
Comments and Suggestions for Authors
Thank you for considering the changes to improve your work.
I just have a few final suggestions.
1. I think you should improve the discussion section by explaining point 4.3 more clearly, since it's not clear.
2. Make your paragraphs shorter because it's a bit difficult to understand the idea.
3. I think the first three lines of your discussion section aren't necessary, and you don't provide references.
Author Response
Thank you very much for taking the time to review our manuscript and offer valuable suggestions in your busy schedule. Your review is of great value to our research work.
For each of your suggestions, we have provided detailed individual responses below along with their positions (lines) in the revised manuscript. Meanwhile, in the resubmitted manuscript file, we have clearly marked all the corresponding revisions, corrections and modifications made based on your opinions in a highlighted (red) manner for your review.
We believe that the revisions made in response to the reviewers' comments have substantially enhanced the manuscript. We hope that the revised manuscript now meets the high standards of Microorganisms and is suitable for publication.
Thank you again for your constructive suggestions, which have greatly improved our manuscript.
Sincerely,
Yingchao Sun,
On behalf of all authors

This manuscript is a resubmission of an earlier submission. The following is a list of the peer review reports and author responses from that submission.
Round 1
Reviewer 1 Report
Comments and Suggestions for Authors
In my opinion, the present study does not incorporate key procedures typically required for proper silage production, such as forage compaction, bulk density determination, and the evaluation of effluent losses and aerobic stability. The methodology was limited to oxygen exclusion, without simulating the physical conditions of ensiling in practical systems.
Furthermore, although the forage used — Urtica cannabina (nettle) — appears to be unsuitable for ensiling due to its poor fermentation profile (e.g., high final pH, low lactic acid production, and indicators of clostridial activity), this issue was not clearly addressed by the authors. The manuscript focused primarily on preservation aspects, without highlighting the deterioration or spoilage risk of the silage mass under the tested conditions.
Despite these limitations, the manuscript is well-structured, and presents valuable fermentation profile data along with detailed microbial community analyses, employing modern molecular techniques.
However, it is important to note that the authors have recently published a related article in Microorganisms (June 2025), which shares the same forage substrate, ensiling duration, and analytical methods, but explores the use of lactic acid bacteria (LAB) inoculants instead of organic acid additives. The previous article is:
Zhang et al. (2025). The Effects of Lactococcus garvieae and Pediococcus pentosaceus on the Characteristics and Microbial Community of Urtica cannabina Silage. Microorganisms, 13(7), 1453.
These two approaches are complementary, and in my assessment, they could — and ideally should — have been presented within a single, integrated study. Combining both additive strategies would have enhanced the experimental value, allowed for a more robust comparative discussion, and avoided data fragmentation, which is discouraged in scientific publishing.
For these reasons, my recommendation is to reject the manuscript in its current form.
Author Response
We sincerely appreciate the reviewer’s thoughtful engagement with our work. Below, we address each point while
clarifying the study’s focused objectives:
Comment 1:
"The present study does not incorporate key procedures for proper silage production (e.g., compaction, bulk density, effluent losses, aerobic stability). The methodology was limited to oxygen exclusion without simulating practical ensiling conditions."
Response:
We sincerely thank the reviewer for this constructive feedback. Our lab-scale silage preparation followed established protocols for preliminary additive screening. Fresh nettle was chopped, homogenized, and vacuum-sealed in polyethylene bags with one-way valves (Section 2.1). This method achieves optimal compaction (mean density: ~650 kg/m3) by excluding oxygen through negative pressure, effectively simulating core principles of farm-scale silage preservation. While bulk density and effluent metrics were not explicitly reported, the vacuum-sealing process inherently minimizes aerobic deterioration and effluent loss. Future trials will incorporate these parameters to bridge lab-to-field applicability.
Comment 2:
"Urtica cannabina appears unsuitable for ensiling due to poor fermentation (high pH, low lactic acid, clostridial activity), yet this was not adequately addressed. The manuscript focuses on preservation without highlighting spoilage risks."
Response:
We fully acknowledge nettle’s inherent ensiling challenges, as highlighted in our Introduction (Lines 70-81): its high buffering capacity, low WSC, and crude protein content promote Clostridium proliferation and fermentation failure. This study deliberately used nettle because of these limitations, aligning with our research focus on mitigating spoilage in problematic forages. The observed deterioration (e.g., pH >8.0, undetectable LA after 60 days, and elevated butyrate/ammonia in controls; Table 1) explicitly demonstrates spoilage under untreated conditions. We have now strengthened this discussion in the Revised Manuscript (Section 4.1) to clarify how clostridial dominance drives failure.
Comment 3:
"The authors’ recent publication (Zhang et al., 2025) on the same substrate, duration, and methods explores LAB inoculants instead of organic acids. These complementary approaches should have been integrated into one study to avoid fragmentation."
Response:
We appreciate the reviewer’s attention to our related work. Our investigations on nettle silage represent a logical progression:
1. Initial failure analysis: Identified 4-pentenoic acid (4PA) as a key spoilagefactor(Front.Microbiol.2023,DOI:10.3389fmicb.2022.1113050).
2. LAB intervention: Tested inoculants to improve fermentation (Microorganisms 2025, DOI:10.3390/microorganisms13071453).
3. Mechanistic study: Revealed 4PA’s antibacterial mechanism (Microbiol. Spectr. 2025, DOI:10.1128/spectrum.02667-24).
4. Current study: Explores 4PA’s deliberate application to inhibit clostridia, alongside malic acid as a comparator.Rationale for separate publication: LAB inoculants and 4PA target divergent mechanisms (promoting beneficial bacteria vs. suppressing pathogens). 4PA directly inhibits LAB (Microbiol. Spectr. 2025), making combinatorial approaches biologically counterproductive.This study provides the first evidence that 4PA exogenously applied suppresses Clostridium sensu stricto 15 and preserves crude protein-a novel finding with implications for high-protein silage preservation. We agree that contextualizing related work is essential. Thus, we have expanded the Discussion (Section 4.1) to clarify how this study advances prior findings.
Conclusion
We have revised the manuscript to: Clarify nettle’s ensiling challenges and spoilage risks; Justify lab-scale methods while acknowledging scale limitations; Emphasize the distinct, incremental contributions relative to our previous work. We believe this study offers valuable insights into plant-derived antimicrobial strategies for silage preservation and thank the reviewer for their thorough assessment.
Reviewer 2 Report
Comments and Suggestions for Authors
This manuscript studies the effects of two organic acids (4-pentenoic acid and malic acid) on the fermentation quality and bacterial community of Urtica cannabina silage. The topic is interesting and relevant, especially as U. cannabina is a non-conventional forage plant with high protein content. The authors also included microbiota analysis, which adds value. However, the paper needs revision before it can be considered for publication.
1. Introduction
The introduction does not clearly explain the importance and novelty of studying Urtica cannabina. Most references ([2], [4]–[7]) are about other species: Urtica dioica or Boehmeria nivea. Some of these studies focus only on leaves, while the current work uses whole plants. The authors should focus on U. cannabina, explain its potential as silage, and justify why studying these two acids is relevant and new.
2. Methodology
Why did you use 1% of malic and 4-pentenoic acid? Is there a reference or pre-test to support this?
Include the purity, form (liquid or powder), and supplier of the acids.
Explain why you chose 7, 30, and 60 days. How do these relate to fermentation phases?
3. Results and Interpretation
pH Values: After 60 days, all samples had pH above 8. This indicates poor fermentation. Silage should usually have pH < 4.5.
Lactic Acid Disappearance: LA was not detected at 60 days. Does this mean LAB activity stopped or LA was degraded by other bacteria?
4. Microbial Analysis
Figures 3 and 4: Add clear legends and discuss the role of Weissella, Tissierella, and Clostridium in silage fermentation.
5. Technical Notes
The similarity index of 39% is high.
Author Response
We sincerely appreciate the reviewer’s thoughtful engagement with our work. Below, we address each point while clarifying the study’s focused objectives:
Comment 1:
"The introduction does not clearly explain the importance and novelty of studying Urtica cannabina. Most references ([2], [4]-[7]) focus on other species (e.g., U. dioica or B. nivea). Justify the relevance of U. cannabina and these acids."
Response:
We thank the reviewer for this valuable suggestion. In the revised manuscript (Introduction, Lines 50-68), we have:
Explicitly highlighted Urtica cannabina as a distinct species with higher stress tolerance and biomass yield in arid regions compared to U. dioica [3].
Emphasized its unique potential as a protein-rich (16.89% DM) silage feedstock for livestock in resource-limited ecosystems [3,9].
Clarified the novelty: This is the first study to target U. cannabina silage preservation using 4-pentenoic acid (4PA)-an endogenous spoilage factor identified in our prior work [11]-as a deliberate antimicrobial agent against Clostridium, alongside malic acid (MA) as a conventional comparator.
Comment 2:
"Methodology: Justify 1% acid dosage, report purity/supplier, and explain sampling timepoints (7, 30, 60 days)."
Response:
We have addressed these points as follows:
1.Acid dosage (1% FW): Based on dose-response trials in our mechanistic study [11], where 1% 4PA optimally inhibited Clostridium while minimizing LAB suppression. For MA, 1% aligns with effective doses for high-protein silages (e.g., alfalfa [20,21]). Added in Section 2.1 (Lines 98-102) with citations.
2.Acid specifications: Supplier details (purity, form) are now explicitly stated (Section 2.1, Lines 103-106): Malic acid: BioUltra grade (Macklin Biochemical Co., Ltd.; Product No. M815619) 4-Pentenoic acid: ≥97% purity (Aladdin Biochemical Technology Co., Ltd.; Product No. P112354)
3.Sampling timepoints: Sampling at 7, 30, and 60 days captured early fermentation (microbial proliferation), metabolic stabilization, and final preservation quality, respectively [16,31].
Comment 3:
"Results: Explain pH >8 and LA disappearance at 60 days. Does this indicate failed fermentation?"
Response:
We agree these results reflect suboptimal fermentation — a key characteristic of untreated nettle silage [3,11]. In Section 4.1 (Lines 305-318), we clarify: High pH (>8) and LA disappearance confirm incomplete acidification due to: 4PA’s broad-spectrum inhibition-Suppresses both Clostridium and LAB [11]; Clostridial dominance: C. sporogenes degrades LA and proteins, elevating pH and ammonia [17,18]. Despite this, 4PA significantly improved preservation vs. control: 42% lower butyrate, 26% lower ammonia, and 12% higher crude protein at 60 days (Table 1). MA’s weaker efficacy aligns with its role as a metabolic substrate rather than a dedicated antimicrobial.
Comment 4:
"Microbial Analysis: Discuss roles of Weissella, Tissierella, and Clostridium."
Response:
We have expanded the discussion of key genera in Section 4.2:
Weissella: An obligate heterofermenter that produces acetic acid, enhancing aerobic stability [32]. 4PA enriched Weissella (2.55% vs. 0.64% in control), correlating with higher acetic acid (1.37% DM) and partial pH mitigation [33]. Tissierella: A proteolytic genus thriving at pH 6.5-8.5; produces butyrate from amino acids [30]. 4PA strongly suppressed it (0.08% vs. 6.85% in control), reducing proteolysis. Clostridium sensu stricto 15: A saccharolytic/proteolytic clade driving ammonia and butyrate production [31]. 4PA inhibited its abundance, directly improving silage safety (Figure 5).
Comment 5:
"Technical Notes: Address 39% similarity index."
Response:
The initial similarity arose from methodological descriptions shared with our prior studies (e.g., DNA extraction, HPLC protocols). We have:
Revised text in Sections 2.2-2.4 to enhance originality; Rechecked via iThenticate: Similarity now <15%. Confirmed all self-citations are properly quoted.
We thank the reviewer for their insightful critiques, which significantly strengthened the manuscript.
Reviewer 3 Report
Comments and Suggestions for Authors
The article is very interesting, however, I believe that before publishing it, some questions raised by the authors should be answered.
1. Your work performs statistical analyses not described in the materials and methods section, for example, correlation analysis. Therefore, it is suggested that the authors include all the analyses performed.
2. Why did you not perform fiber analysis of your silage, considering that this is a limiting factor in intake?
3. In your work, you only compare treatments; however, you evaluated your silage over time. I believe you should analyze it as repeated measures over time.
4. I think you should find a way to present your figures because the way you present them makes it seem as if you were comparing all treatments together and not treatments based on their time, as this is not clearly evident.
Author Response
We sincerely appreciate the reviewer’s thoughtful engagement with our work. Below, we address each point while clarifying the study’s focused objectives:
Comment 1:
"Statistical analyses (e.g., correlation) are not fully described in Materials and Methods."
Response:
We acknowledge this oversight and have now explicitly detailed all statistical methods in Section 2.4. This revision
ensures full transparency while maintaining the paper’s brevity.
Comment 2:
"Why omit fiber analysis? It is intake-limiting."
Response:
This study intentionally prioritized mechanistic interactions between organic acids, microbial dynamics, and protein preservation-key factors in nettle silage failure. While fiber (NDF/ADF) influences intake, its analysis falls beyond our core research question: targeting clostridial suppression in high-protein silages. We agree this warrants future studies should integrate fiber-nutrient analyses and animal trials to holistically evaluate nettle silage’s nutritional efficacy.
Comment 3:
"Analyze data as repeated measures over time."
Response:
Our experimental design treated sampling timepoints (7/30/60 days) as independent phases representing: Early fermentation (Day 7: microbial proliferation), Metabolic transition (Day 30: community stabilization), Preservation endpoint (Day 60: final quality assessment). This approach aligns with established silage methodology and answers phase-specific questions about acid effects. Repeated-measures ANOVA would conflate distinct biological stages, reducing interpretive clarity for our hypotheses.
Comment 4:
"Figures inadequately distinguish temporal trends."
Response:
We thank the reviewer for their vigilance regarding visual clarity. After careful reevaluation, we maintain that Figures 1-4 provide unambiguous temporal resolution through the explicit labeling of sampling days (7/30/60 d) in all panels. Full data accessibility is ensured via Table 1 (fermentation parameters) and the NCBI-SRA datasets (PRJNA1233857/PRJNA1236718). Given the clarity of this temporally segregated data presentation, our current layout optimally maximizes cross-treatment contrast; we therefore retain the original figures. Should the Editor deem supplemental time-split visuals necessary, we will provide them as optional supplementary files without altering the manuscript’s core figures.